# CatLC: Catalonia Multiresolution Land Cover Dataset

**Carlos García Rodríguez**
Institut Cartogràfic i
Geològic de Catalunya &
Universitat de Barcelona
c.garcia.r@icgc.cat

**Oscar Mora**
Institut Cartogràfic i
Geològic de Catalunya
oscar.mora@icgc.cat

**Fernando Pérez-Aragüés**
Institut Cartogràfic i
Geològic de Catalunya
fernando.perez@icgc.cat

**Jordi Vitrià**
Universitat de
Barcelona
jordi.vitria@ub.edu

## Abstract

Traditional natural image datasets are very rich. However, only a few remote sensing datasets are available and cover a tiny territory or cover a larger one with low spatial resolution and/or few classes. In this paper, we present the *Catalonia Multiresolution Land Cover Dataset* (CatLC), a remote sensing dataset. The dataset contains images at different spatial resolutions captured by both aircraft and satellites (Sentinel-1 and Sentinel-2), in addition to topographic maps. All this dataset has been created with images from the Cartographic and Geological Institute of Catalonia (ICGC) catalogs and the European Space Agency (ESA). The ICGC's land cover ground truth accompanies these images with 41 classes at a spatial resolution of 1 m in an area of 32000 km$^2$, covering the Spanish region of Catalonia. CatLC is a multilayer, multiresolution, multimodal, multitemporal dataset, which has excellent potential for the Artificial Intelligence (AI) community and the exploration of modeling methodologies. Land cover maps are used in different realms such as forestry for inventory area estimates, hydrology regarding microclimatic variables, agriculture to improve irrigation or geology in geohazards, and risk identification and assessment. Therefore, accurate and updated knowledge about land changes is essential for territory management with different purposes over multiple fields. Using various combinations of the images from the dataset, we offer a benchmark that could serve as a starting point to explore artificial intelligence techniques for remote sensing segmentation purposes. In this vein, CatLC dataset aims to engage with computer vision experts interested in remote sensing and stimulate research and development. The CaTLC dataset is available at https://www.icgc.cat/en/Downloads/Aerial-and-satellite-images/Contingut/Catalonia-Multi-resolution-Landcover-Dataset-CatLC.

## 1 Introduction

Pixel-wise, human annotation of satellite images is challenging and dubious and thus often requires manual annotation with tools like Google Street View, fieldwork, etc. Therefore, mapping agencies are on the quest to explore how to substitute their strenuous and time-consuming manual tasks for automated processes. To do so, current accuracy in automatic land cover segmentation requires improvement in terms of methodologies and data coming from airborne and satellite sensors. Land

Submitted to the 35th Conference on Neural Information Processing Systems (NeurIPS 2021) Track on Datasets and Benchmarks. Do not distribute.

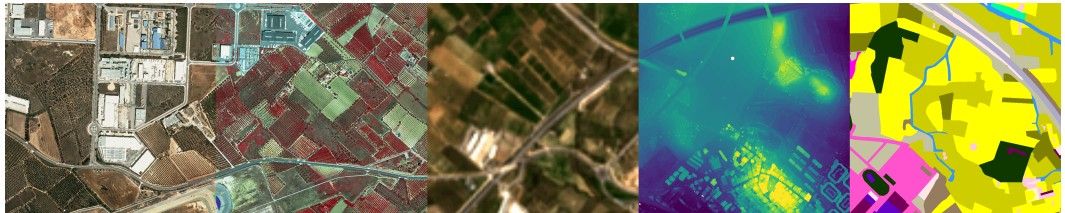

Figure 1: Continuous area using different layers of the dataset together with the ground truth.

cover segmentation is among the primary uses of airborne and satellite images and the proposed focus of this work.

Land cover maps are used in different realms such as forest inventory or forest management, hydrology regarding microclimatic variables, agriculture to improve crop management or geology in risk identification and assessment.Therefore, accurate and updated knowledge about land dynamics is essential for territory management with different purposes and in multiple fields. These land cover maps are provided by institutions that invest time and human resources to fulfill the costly task of producing them.

In cartographic institutions, due to their heritage, legal assessments and administrative framework, land segmentation is still done mainly employing photointerpretation techniques, entailing very high costs in terms of time and human resources. Thus, cartographic institutes are transitioning from manual land segmentation to automation [1]. The transformation towards an automatic solution tends to face a critical point: the scarcity of high-quality datasets. In this regard, we find datasets composed of canonical images such as ImageNet[2] and PASCAL VOC Dataset [3]. Labeling the images of such datasets does not pose an interpretation problem as they are distinctive. However, labeling aerial images correctly might be challenging, i.e., to differentiate between the deciduous or evergreen forest.

In this paper, we present the Catalonia Multiresolution Land Cover Dataset (CatLC). This dataset comprises a large variety of images: orthophoto RGB and infrared from airborne sensors at high resolution, radar from satellite sentinel-1, multispectral from satellite sentinel-2, and composition of topographic maps—all those accompanied by a land cover map minutely labeled by experts in photointerpretation. Using different combinations of the images from the dataset, we offer a benchmark that could serve as a starting point to explore different artificial intelligence techniques for remote sensing segmentation purposes. CatLC dataset aims to engaging with computer vision experts interested in remote sensing and stimulate research and development.

## 2 Previous work

In 2017, a competition was held to find the best automatic labeling algorithm for two German cities, preparing two different datasets [4]. The first one was in Vaihingen, and the dataset was composed of 38 different tiles and consisted of an orthophoto (infrared, red, and green bands) and a Digital Surface Model (DSM). The resolution (ground sampling distance) of the orthophoto and the DSM was 9 cm. The second one in Potsdam consisted of an orthophoto (4 bands including infrared) and a DSM, all at 5 cm spatial resolution. These data were arranged in 38 tiles. Ground truth consisted of six classes: impervious surfaces, buildings, low vegetation, trees, cars, and background. Both datasets had a very high resolution, but they covered a small area.

There is also the Inria (France) dataset that covers an area of 810 km$^2$ comprising an orthophoto layer at a spatial resolution of 30 cm and ground truth consisting of two classes (building/no-building) in several cities with different populations, from downtown San Francisco to alpine cities in Austria [5]. Another example of the use of remote sensing is the SpaceNet Road Network Detection dataset. This dataset has more than 8000 km of roads and four cities: Las Vegas, Paris, Shanghai, and Khartoum. The purpose of this dataset is to find the roads and classify them into 7 different types [6].

In addition, a few years ago, the US Geological Survey released a land cover dataset of the state of New York where we can find 21 classes at a spatial resolution of 30 m per pixel [7]. Finally, DLR (German Aerospace Center) also produced a dataset comprising 31 semantic categories and 12 subcategories of lane markings for different German and European cities. This dataset contains

Table 1: Summary of some of the most used datasets in remote sensing applications for land cover segmentation compared to CatLC. Some datasets do not specify the resolution at which the ground truth was labeled.

| Dataset | Bands | Classes | Tiles | Tile shape (pixels) | Ground truth resolution (m/pixel) |
|---|---|---|---|---|---|
| Vaihingen | 5 | 6 | 38 | $2000 \times 2000$ | unknown |
| Potsdam | 5 | 6 | 38 | $6000 \times 6000$ | unknown |
| Inria | 3 | 2 | 180 | $5000 \times 5000$ | 0.3 |
| USGS New York | - | 21 | 1 | $16989 \times 22610$ | 30 |
| SkyScapes (DLR) | 3 | 31 | 16 | $5616 \times 3744$ | 13 |
| FloodNet | 3 | 10 | $\sim 400$ | $4000 \times 3000$ | unknown |
| CatLC | 21 | 41 | 34040 | $960 \times 960$ | 1 |

16 images of size $5616 \times 3744$ at 13 cm spatial resolution [8]. More recently, we find the Floodnet dataset, which provides high-resolution UAS imageries with detailed semantic annotation [9]. In Table 1 we summarize the description of the aforementioned individual datasets.

# 3    CatLC: Catalonia Land Cover Dataset

In this section, the CatLC dataset is presented in detail, consisting of a set of images obtained by airborne and satellite sensors from the catalogs of the Cartographic and Geological Institute of Catalonia (ICGC) and the European Space Agency (ESA), and the current ICGC's land cover map.

It covers the entire territory of Catalonia (Spain), approximately 32000 km$^2$, providing an extraordinary source of information for the application of AI and Deep Learning (DL) techniques, both regarding the quality and variety of the images and their extension, not limited to a small part of the territory. Catalonia is located on the shores of the Mediterranean Sea and has a great variety of types of land cover, which makes it very suitable to create a heterogeneous dataset with numerous labels.

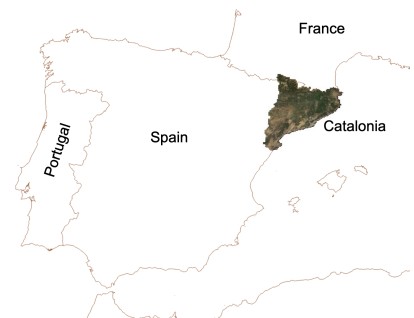

Figure 2: Location of the area of interest, Catalonia (Spain).

All the images have been acquired during 2018 at different spectral bands and spatial resolutions and are provided in GeoTiff raster format. They all share a common georeferencing at WGS84 UTM31N Reference System projection and they cover the same geographic extension, given by the following bounding box: UTM X West: 240000, UTM X East: 540000, UTM Y North: 4780000, and UTM Y South: 4480000. The different types of images, with spatial resolutions that vary between 1 and 10 meters, depending on the product and sensor used, are presented in detail in the following subsections. In such subsections we also illustrate the bands provided by the dataset (land cover, orthophoto, Sentinel-1, Sentinel-2 and topographic maps). The figures 4, 5, 6, 7, 8, 9, 10,11, 12, show the different bands on three geographical areas in Catalonia (area A: 42°08'02.6"N 1°29'30.7"E, area B: 42°02'23.0"N 2°39'56.0"E and area C: 41°57'50.3"N 2°08'42.4"E). We have summarized all available data in Table 3.

## 3.1    Land Cover Map

The 2009 land cover map presented here is a simplification to 41 classes of the Land Cover Map of Catalonia (version 4) generated by CREAF (Center for Ecological Research and Forestry Applications) [10], together with an adaptation to the data model approved by the Cartographic Coordination Commission of Catalonia which allows the comparison between maps of different years.

The 2018 land cover map is an update of the 2009 map, made from the photointerpretation of ICGC's 2018 orthophotos. The minimum area for labeling an element was 500 squared meters and the

Table 2: CatLC Dataset with 41 classes and legend.

| Agricultural area | | |
|---|---|---|
| 01 Herbaceous crops | 14 Meadows and grasslands | 28 Sports and leisure areas |
| 02 Orchard, plant nurseries... | 15 Shore forest | 29 Mining or landfills |
| 03 Vineyards | 16 Bare forest soil | 30 Areas in transformation |
| 04 Olive groves | 17 Burned areas | 31 Road network |
| 05 Other woody crops | 18 Rocky | 32 Urban bare ground |
| 06 Crops in transformation | 19 Beaches | 33 Airport areas |
| **Forest area** | 20 Wetlands | 34 Railway network |
| 07 Dense coniferous forests | **Urban area** | 35 Port areas |
| 08 Dense deciduous forests | 21 Urban area | **Water bodies** |
| 09 Dense forests of sclerophylls | 22 "Eixample" | 36 Reservoir |
| 10 Scrub | 23 Lax Urban Areas | 37 Lakes and lagoons |
| 11 Clear coniferous forests | 24 Isolated buildings | 38 Watercourses |
| 12 Clear deciduous forests | 25 Isolated residential areas | 39 Rafts |
| 13 Clear forests of sclerophylls | 26 Green areas | 40 Artificial channels |
| | 27 Industrial or commercial | 41 Sea |

minimum length for linear features such as roads, rivers, railroad tracks, etc. was between 8 and 10 meters. The working scale for the photo-interpreters has been 1:2500. The changes bigger than 2 ha in extent have been identified using the following information:

- Comparison of LANDSAT images of both years, 2009 and 2018.
- The changes reflected in ICGC's 1: 5000 Topographic Base.
- The forest fires and the SIGPAC (Geographic Information System of Agricultural Plots) data for this period.

The change layer between 2012 and 2018 of the Corine Land Cover has also been considered and special attention has been paid to the categories of crops in transformation and zones in transformation of the 2009 version due to their greater dynamics.

Exceptionally, due to the large volume of changes in the agricultural areas of Lleida and the coastal area from Tarragona to Barcelona, only agricultural changes of more than 5 hectares have been photo-interpreted. Supervision has been performed on a sample of 811 points throughout this territory, resulting in a thematic accuracy of 81%. The final 41-class (see Table 2) product presented in this publication is delivered at spatial resolution of 1m. The distribution of the land covers along the territory is heterogeneous. Some covers as herbaceous crops or dense coniferous forests are way more common than airport areas or rafts. In Figure 3, we can see the histogram for the complete dataset.

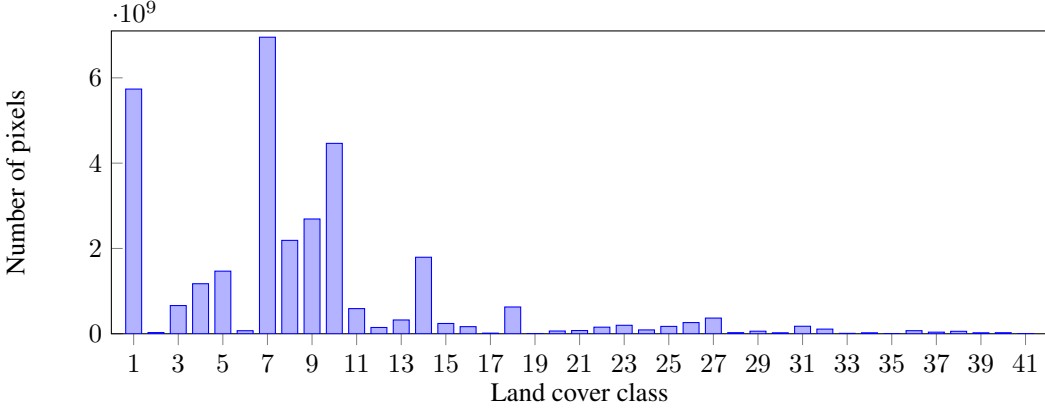

Figure 3: Class distribution on CatLC dataset.

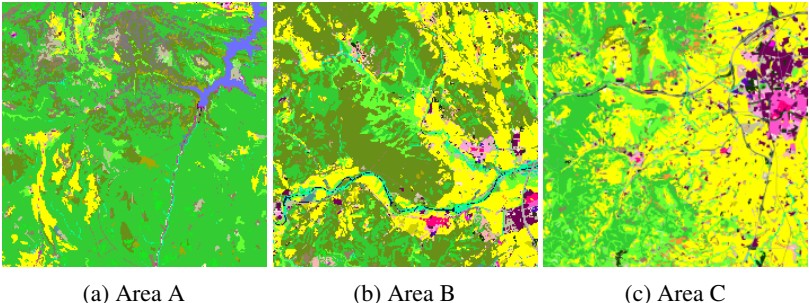

(a) Area A        (b) Area B        (c) Area C

Figure 4: Land cover map samples.

## 3.2 Orthophoto

An orthophoto is a cartographic document consisting of a vertical aerial image that has been rectified in such a way as to maintain a uniform scale over the entire image surface. It consists of a geometric representation at a given scale of the Earth's surface. The original images were taken with a resolution of 25 centimeters ([https://www.icgc.cat/en/Downloads/Aerial-and-satellite-images/Conventional-orthophoto](https://www.icgc.cat/en/Downloads/Aerial-and-satellite-images/Conventional-orthophoto)), but because the land cover map has a resolution of 1 meter, we have decided to rescale the orthophoto raster layer also to 1 meter.

This layer comprises four distinct bands, each providing information from different zones of the electromagnetic spectrum. Three of them belong to the visible area of the spectrum (RGB) and one of them to the infrared area. On this cartographic document, digital make-up tasks have been carried out to minimize artifacts that may have originated during the generation process or the acquisition of the images. This way, it can be assured that the affected area does not exceed 1% of the total area of Catalonia.

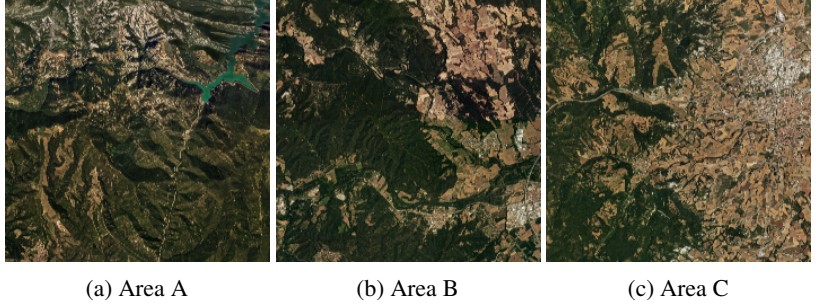

(a) Area A        (b) Area B        (c) Area C

Figure 5: Orthophoto RGB samples.

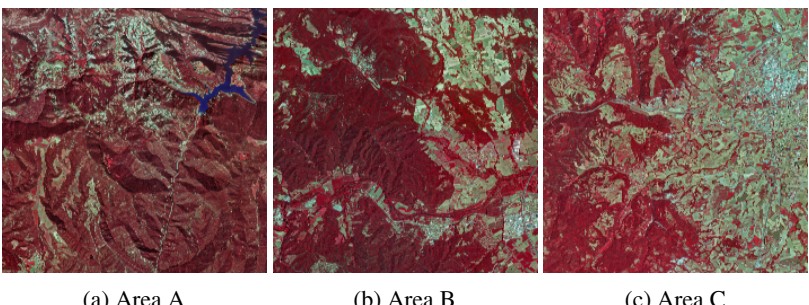

(a) Area A        (b) Area B        (c) Area C

Figure 6: Orthophoto (Infrared,R,G) samples.

### 3.3 Sentinel-1

The Sentinel-1 dataset has been generated from radar images (Synthetic Aperture Radar, SAR) in GRD (Ground Range Detected) mode from the year 2018 at 10-meter spatial resolution. The Sentinel-1 satellite constellation is made up of two twin satellites, A and B, from the European Space Agency (ESA). These satellites emit a microwave signal (frequency 5.405 GHz) and subsequently receive the echo of the reflection on the ground surface. Therefore, Sentinel-1 images contain information on the reflectivity of the terrain, which depending on its type (urban, vegetation, crops, water, etc.) will have a different intensity, thus providing valuable information for its classification. For this purpose, 12 acquisitions have been chosen, one for each month of the year, covering the entire territory of Catalonia. Full coverage has been achieved by combining 2 orbits in ascending mode (orbits 30 and 132) and VV (Vertical-Vertical) polarizations in similar dates. The descending orbit and VH (Vertical-Horizontal) polarization have not been included in the present dataset because the information is mostly redundant. However, its use can be explored in case it provides improvements in segmentation. Additionally, an average image of the year 2018 has been generated with improved radiometry (multitemporal speckle reduction) by combining all 12 monthly images into one. Consequently, the average image cannot provide information on temporary changes during 2018 providing however a lower noise level. In appendix B we explain how the Sentinel-1 images are generated.

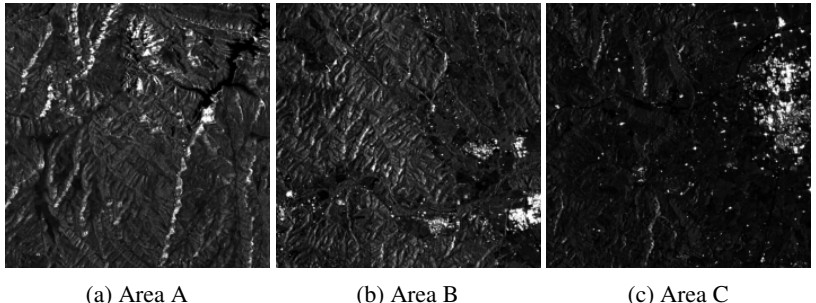

(a) Area A        (b) Area B        (c) Area C

Figure 7: Sentinel-1 (average image during 2018) samples.

### 3.4 Sentinel-2

Sentinel-2 provides multispectral imagery data at different resolutions approximately every five days. We have selected two relevant dates for this dataset, the first one in April and the second one in August. We have chosen these two dates to follow the phenological evolution of the vegetation throughout the spring and late summer. As we are in the Mediterranean area, with these two dates it is possible to detect both winter and summer herbaceous crops as well as evergreen and deciduous forest masses. Due to the presence of clouds, multiple data takes have been necessary to make a cloud-free mosaic. In appendix B we explain how the Sentinel-2 images are generated.

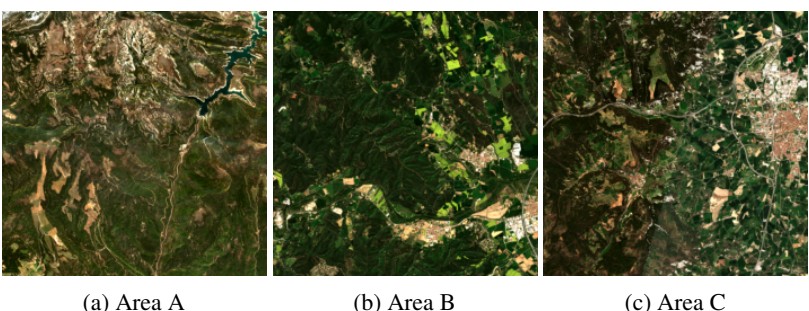

(a) Area A        (b) Area B        (c) Area C

Figure 8: Sentinel-2 RGB (April 2018) samples.

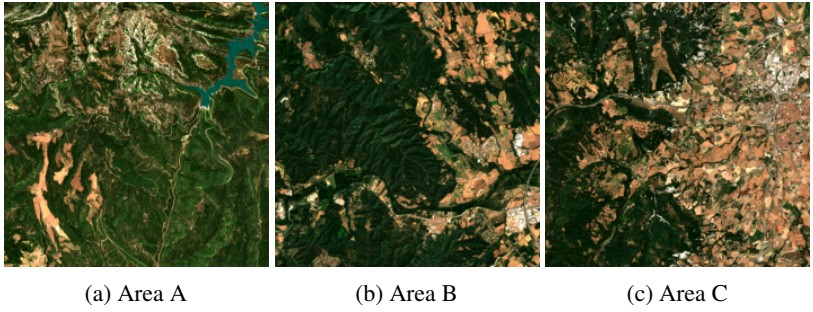

(a) Area A        (b) Area B        (c) Area C

Figure 9: Sentinel-2 RGB (August 2018) samples.

### 3.5 Topographic Maps

Three different topographic products and two subproducts have been generated. Their characteristics are outlined in the following paragraphs.

### 3.5.1 Digital Elevation Model

This is a standard layer freely distributed by ICGC and is built upon the altimetric information of the Topographic Base of Catalonia 1:5000 version 2 (BT-5m v2.0) that includes profiles, altimetric coordinates, breaklines and contour lines, all of them obtained from the terrain. It consists of a raster image at 5m pixel size and its estimated altimetric accuracy is 0.9m RMS.

Two typical subproducts for remote sensing applications are the slope, which indicates each pixel's steepness, and the aspect that indicates the orientation of the maximum slope between adjacent pixels. These values have been calculated from the Digital Elevation Model. Therefore they contain redundant information. We include them because they might be helpful for the interpretation of the results.

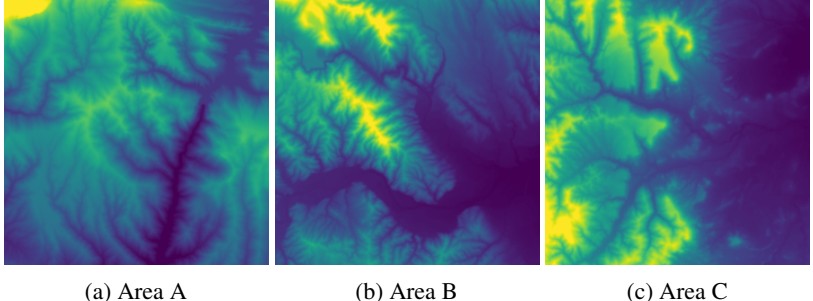

(a) Area A        (b) Area B        (c) Area C

Figure 10: Digital Elevation Model (DEM) samples.

### 3.5.2 Digital Surface Model

This is a raster layer at 1m pixel size containing orthometric heights obtained by automatic correlation between aerial photogrammetric images at 0.25m - 0.35m. It represents the topmost height for every pixel position on the grid be it the ground or, mainly, forest canopy and buildings.

### 3.5.3 Canopy Height Model

The Canopy Height Model (CHM) is a high resolution (1m) raster dataset that maps all the objects over the terrain as a continuous surface. It is advantageous to delineate the forest extent, but it also includes urban landscape data. Each pixel of this model represents the height of the trees above the ground topography. In urban areas, the CHM represents the height of buildings or other built objects.

This layer is built through subtraction of the 2016-2017 LIDAR Digital Elevation Model (`https://www.icgc.cat/en/Downloads/Elevations/Lidar-data`) from the 2018 photogrammetric

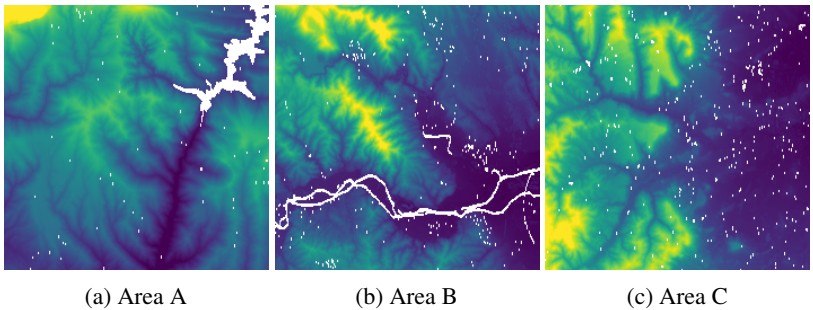

| (a) Area A | (b) Area B | (c) Area C |

Figure 11: Digital Surface Model (DSM) samples.

Digital Surface Model. To clarify, this product is not dependent on the previous Digital Elevation Model. This DEM is produced by LIDAR and the one in the dataset by aerial images.

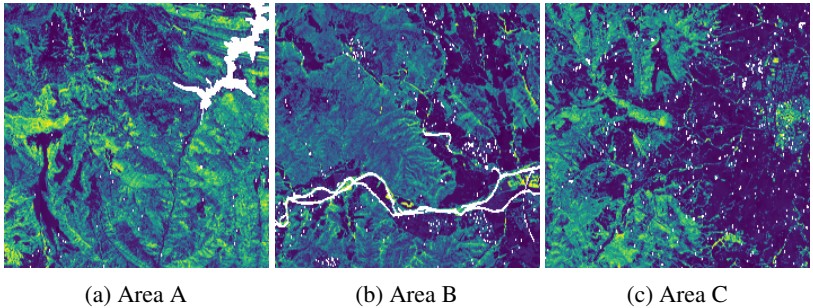

| (a) Area A | (b) Area B | (c) Area C |

Figure 12: Canopy Height Model (CHM) samples.

Table 3: Summary of CatLC dataset.

| Data | Independent bands | Resolution (m) |
|---|---|---|
| Orthophoto | 4 | 1 |
| Sentinel-1 | 12 | 10 |
| Sentinel-2 | 20 | 10 |
| Digital Elevation Model | 1 | 5 |
| Digital Surface Model | 1 | 1 |
| Canopy Height Model | 1 | 1 |
| Land Cover Map | - | 1 |
| 34040 images at $960 \times 960$ pixels per source of 1m data | | |

## 4 Experiments

An initial benchmark accompanies the CatLC dataset to have a starting point and show a helpful pipeline to train a model with raster images. Unlike other datasets that have multiple images, CatLC has only one big image. To work with it, we will need to access smaller tiles, so the first step has been to create a list with the indexes of all the tiles that we are going to use in the dataset of dimension $960 \times 960$ pixels (in the higher spatial resolution images of 1 m). This list was then randomly divided into three groups, 60% for training, 20% for validation, and 20% for testing. Being this is a segmentation problem, we have not been able to have a homogeneous distribution of the three groups because usually, tiles contain multiple classes. The distribution for the sets can be found in Annex C.

As a first use case, we have selected the classical U-Net neural network the [11], which is used as a starting point in most applications that require image segmentation. The experiments are implemented in PyTorch, running in a workstation with a Nvidia Quadro P5000 GPU. Cross entropy loss has been used, together with a commonly used Adam optimizer with 0.0001 as the learning rate.

The experiments have considered three different scenarios:

1. Use as input data orthophoto RGB and infrared. We know by experience that the high resolution should give good results in the frontier of the classes, but its lack of more spectral bands makes it harder to differentiate classes that belong to the agricultural or forest superclasses.

2. Use as input data Sentinel-2 with both months. This time, the low resolution will penalize the frontiers, but there should be an improvement in differentiating agricultural or forest superclasses.

3. It does not make sense to use Sentinel-1 or topographical data all alone because most of its information is about elevation or reflectivity. But combining those, with the orthophoto and Sentinel-2 the results should improve.

To better visualize the results, we have compressed the data in a four superclasses confusion matrix (Figure 13) and mean intersection over union metrics (Figure 15) as recommended in COCO dataset [12]. In Appendix D there is confusion matrix and a mean intersection over union for all the 41 classes.

As we stated before, Sentinel-2 outperforms the orthophoto in agricultural and forest zones, but it loses when we need more resolution as in urban areas. Finally, using the complete dataset gives better results overall. In Figure 14 there is an example of a segmentation using the complete CatLC dataset.

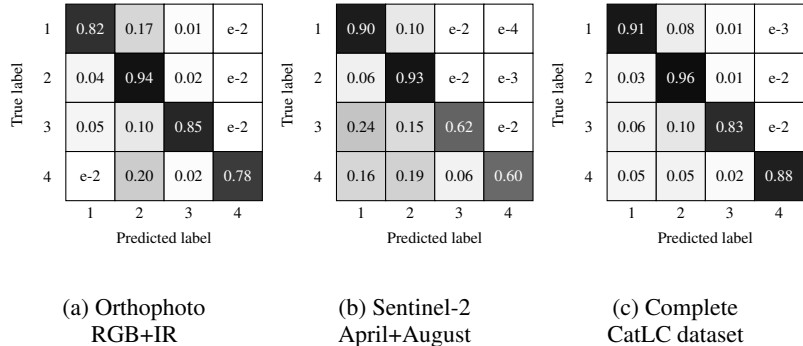

(a) Orthophoto
RGB+IR

(b) Sentinel-2
April+August

(c) Complete
CatLC dataset

Figure 13: Confusion matrix using different input data. All trained with U-Net neural network. The 41 classes have been compacted to the 4 superclasses (1: agriculture, 2: forest, 3: urban, 4: water).

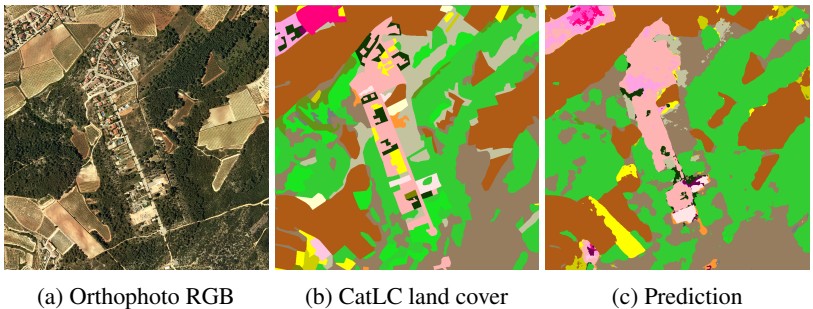

(a) Orthophoto RGB

(b) CatLC land cover

(c) Prediction

Figure 14: Example of U-Net segmentation.

## 5 Discussion

The CatLC dataset contains imagery acquired by airborne and satellite sensors at different spectral bands and spatial resolutions, making it an extraordinary dataset for different AI and DL applications based on remote sensing data. In the dataset the different types of images are presented with detailed information on their technical characteristics, and various tutorials are available to facilitate their use. The volume of data offered is important, approximately 1.4 TB, but the quantity and quality

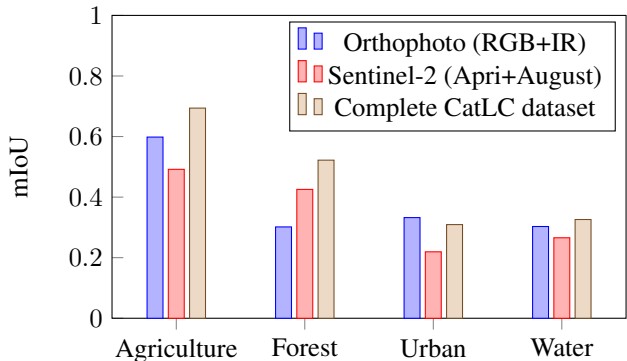

Figure 15: Mean Intersection over Union using different input data for 4 superclasses.

of information offered justifies this volume, being a set of remote sensing data open to the public with characteristics never seen so far in the AI scientific community. Together with the data, a classification study is presented using only a U-Net architecture and orthophoto images (RGB and infrared bands). This example can serve as the first step for more complex studies using other architectures and the entire set of images provided in CatLC. Studies that can determine for example, what type of architectures are more suitable for classifying land covers, and what type of images provide more information and which ones can be discarded due to their null or low contribution to the final result.

CatLC offers images from ICGC's airborne sensors and ESA Sentinel satellites, which have been treated both geometrically and radiometrically for ease of use, even for non-expert users in remote sensing applications. Therefore, the purpose of this publication is to open this type of data to the expert scientific community in AI so that they can analyze the data, develop new methodologies and share their results. The benefits of these investigations will result in automation of production processes that are currently carried out almost manually, as is the case of the land cover mapping, and in the possibility of updating products almost continuously, only depending on the availability of images. This last point should not represent an obstacle in the future, as satellites with remote sensing sensors that continuously monitor the Earth are increasingly available. Finally, it should be noted that CatLC, although focused on creating the land cover map, is open to all kinds of applications related to territory management.

## Acknowledgments

This research was funded by industrial doctorate grant 2018-DI-0045 of AGAUR between University of Barcelona and Cartographic and Geological Institute of Catalonia and RTI2018-095232-B-C21, 2017SGR1742 grants. We want also to acknowledge the work of photointerpreters at the Earth Observation Area (CSPCOT) at Cartographic and Geological Institute of Catalonia, and thank Anna Tardà and Vicenç Palà for their support regarding Sentinel-2 images.

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

# A  Appendix

CatLC is available for download, along with all the necessary information and tutorials, on the following website: `https://www.icgc.cat/en/Downloads/Aerial-and-satellite-images/` `Contingut/Catalonia-Multi-resolution-Landcover-Dataset-CatLC`

There is a tutorial on how to manage the data and visualize it in the following website: `https:` `//github.com/OpenICGC/CatLC/`. There is also the code to reproduce the train presented in the article. We provide the logs for the hole train that can be visualized using Tensorboard.

The use of data is subject to a Creative Common International Recognition 4.0 license. More information. It contains Sentinel Copernicus data modified by the ICGC. It is also requested that the methodologies and results obtained by the different scientific groups are shared with the ICGC through the following e-mail: catlc@icgc.cat.

# B  Preprocessing

**Sentinel-1**

The images were processed with the SNAP (Sentinel Application Platform) software [13] from ESA using the following procedure:

1. Download of the precise orbit for each image using the "Apply-Orbit-File" function, which provides detailed information for its correct georeferencing.
2. Deletion of noisy pixels from the edge of the image using the "Remove-GRD-Border-Noise" function.
3. Radiometric calibration of each image providing calibrated reflectivity information for the Sentinel-1 images. A correct calibration is necessary for the multitemporal study of the data.
4. Topographic effects Compensation using the "Terrain-Flattening" function. The acquisition geometry of the SAR images is oblique, which generates distorting artifacts in the reflectivity associated with the terrain topography (layover, foreshortening and shadowing). This processing compensates for these artifacts to obtain an image that is as independent as possible from the topography.
5. Georeferencing using the "Terrain-Correction" function and final mosaicking of the images.

A video comparing the average Sentinel-1 image and the image for each month is on the CatLC webpage.

**Sentinel-2**

The images obtained by the MSI sensor from the Sentinel-2A and 2B satellites, from the European Commission Copernicus program, have been atmospherically corrected by means of the ESA sen2cor v2.8 software[14] to yield Level-2A images.

The main purpose of sen2cor is to correct single-date Sentinel-2 Level-1C Top-Of-Atmosphere (TOA) radiance from the effects of the atmosphere in order to deliver a Level-2A Bottom-Of-Atmosphere (BOA) reflectance. The process may optionally use a DEM (Digital Elevation Model) to correct the changes in the radiometry related to the topographic relief. A 10m gridded DEM generated at ICGC by photogrammetric techniques has been used in this study. A total of 10 bands, at 10m and 20m resolution, are preserved as input features for the Deep Learning process. Figure 16 presents Sentinel-2 images before and after they have been corrected.

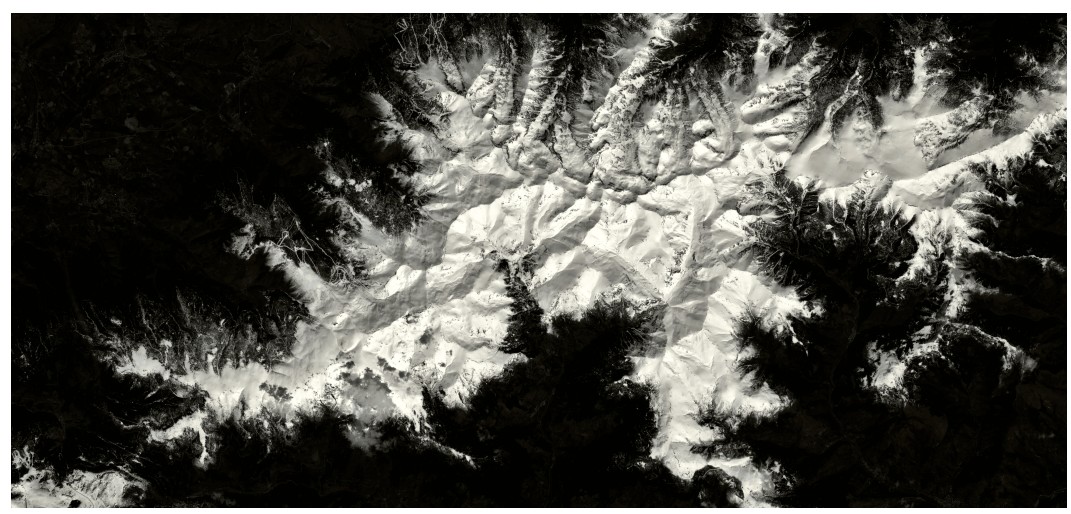

(a) Sentinel-2 original image.

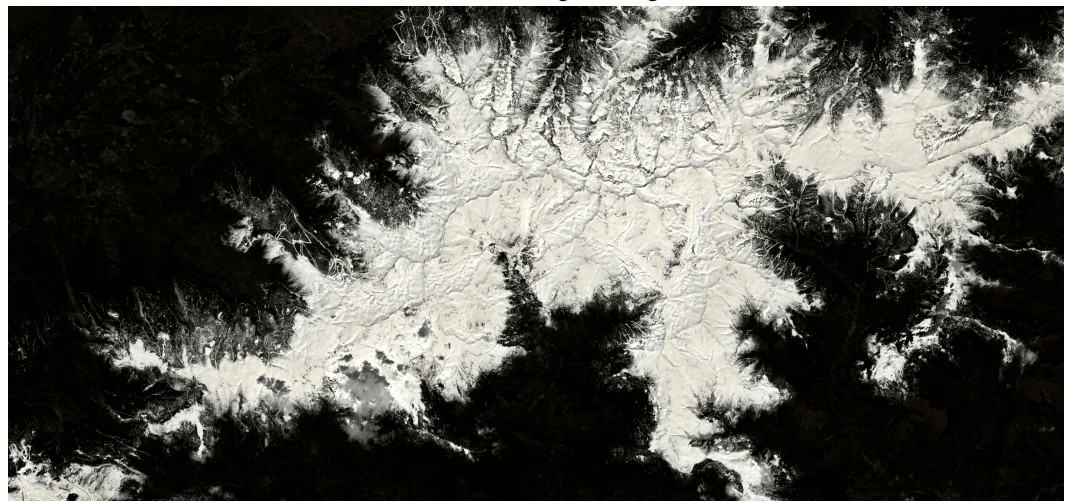

(b) Sentinel-2 after corrections.

Figure 16: Sentinel-2 process with atmospheric and topographic corrections.

 # C Data distribution

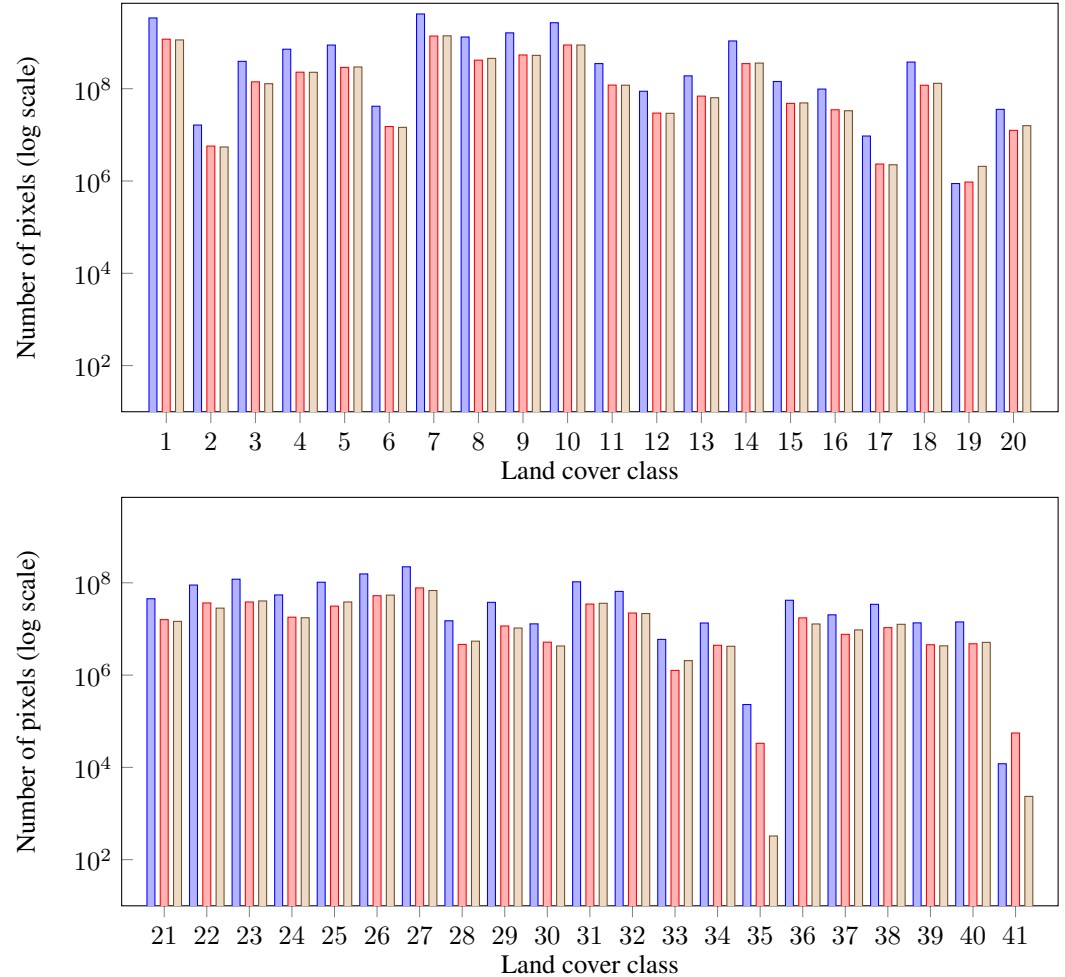

Figure 17: Distribution of the CatLC dataset in three sets: Blue for the train set, red for the validation set and brown for the test set.

 **D   Metrics**

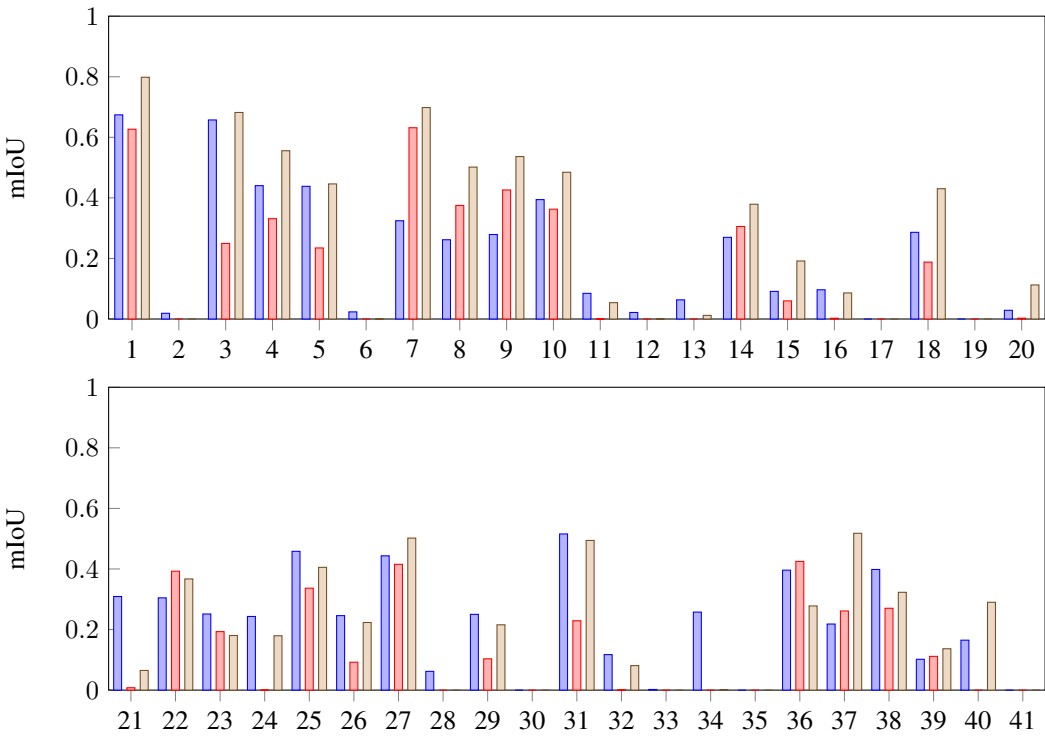

Figure 18: Mean Intersection over Union using different input data for 41 classes.

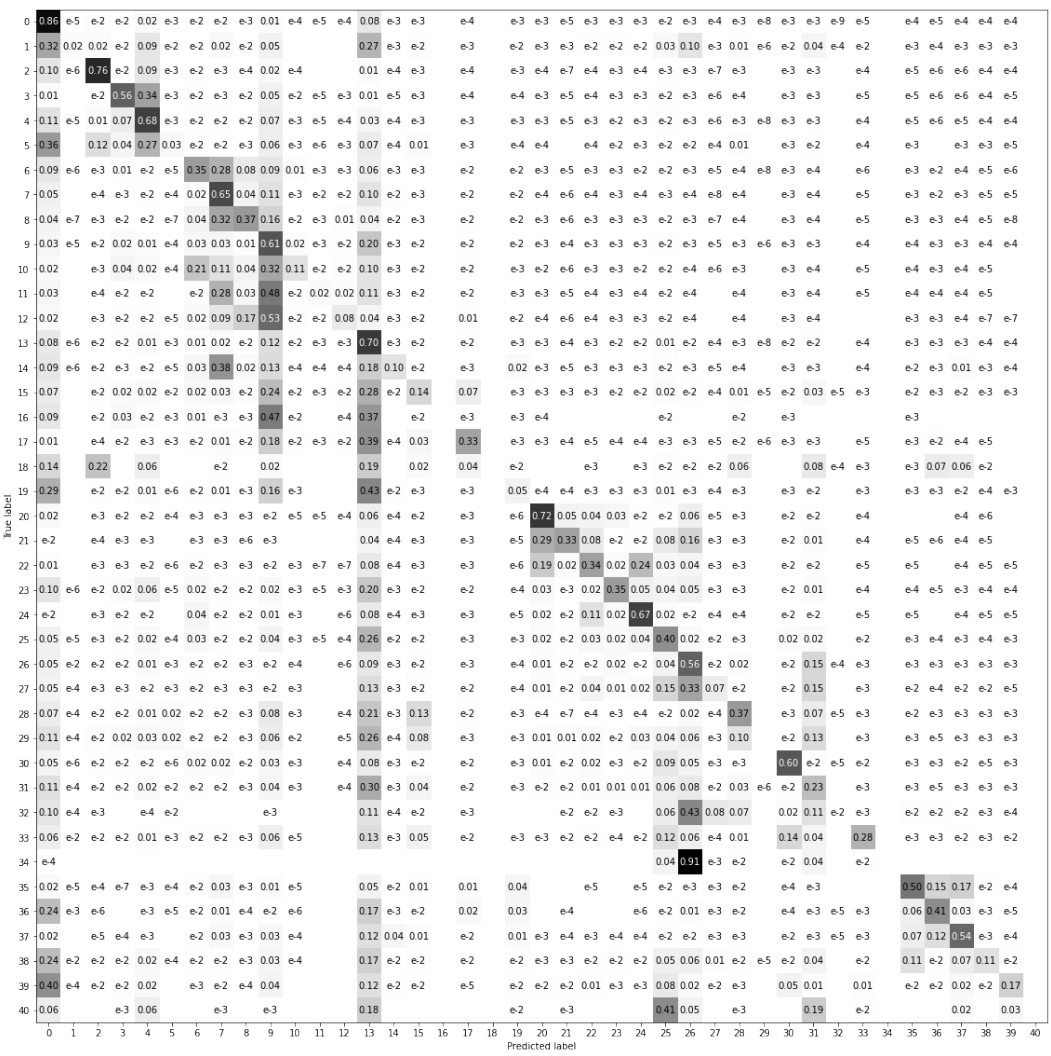

Figure 19: Confusion matrix using the orthophoto (RGB-IR) as input data.

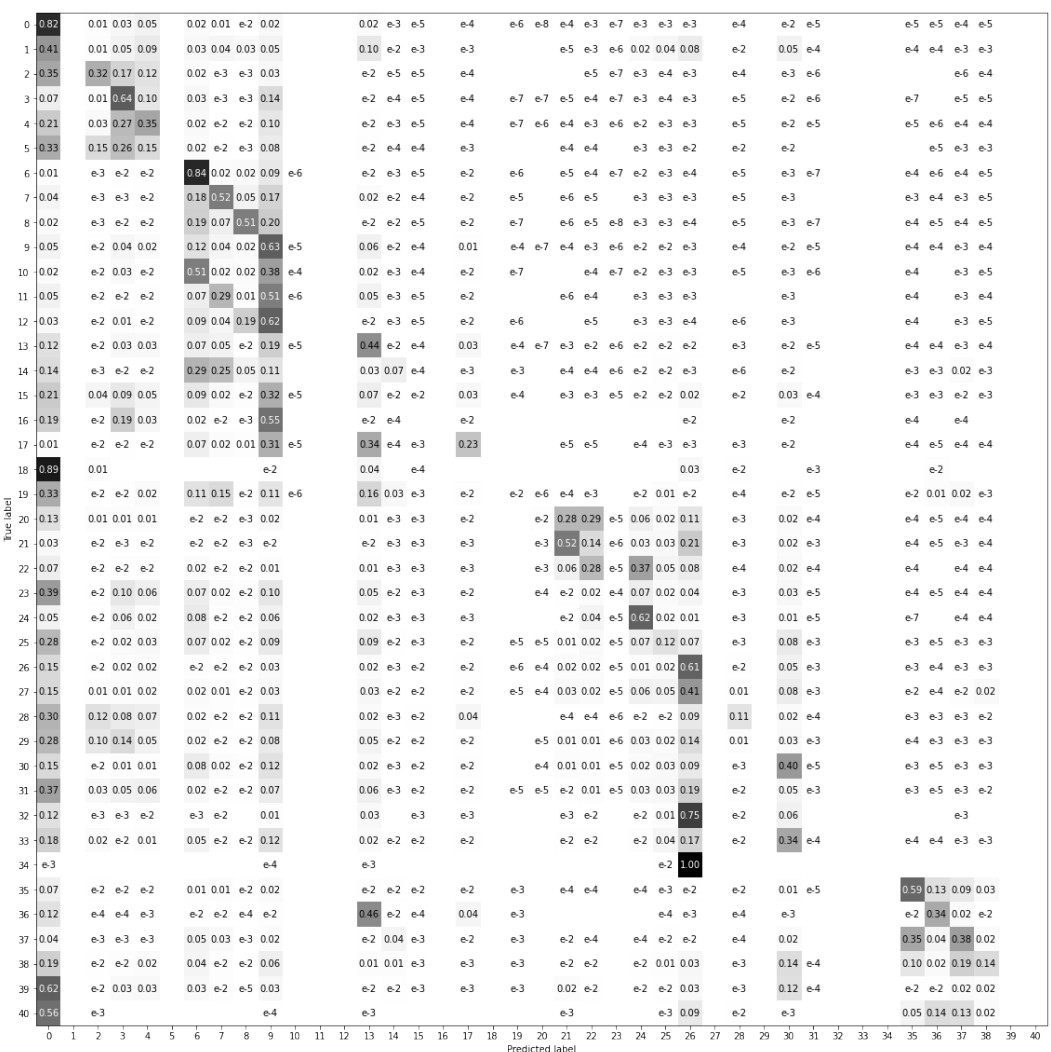

Figure 20: Confusion matrix using Sentinel-2 (April+August) as input data.

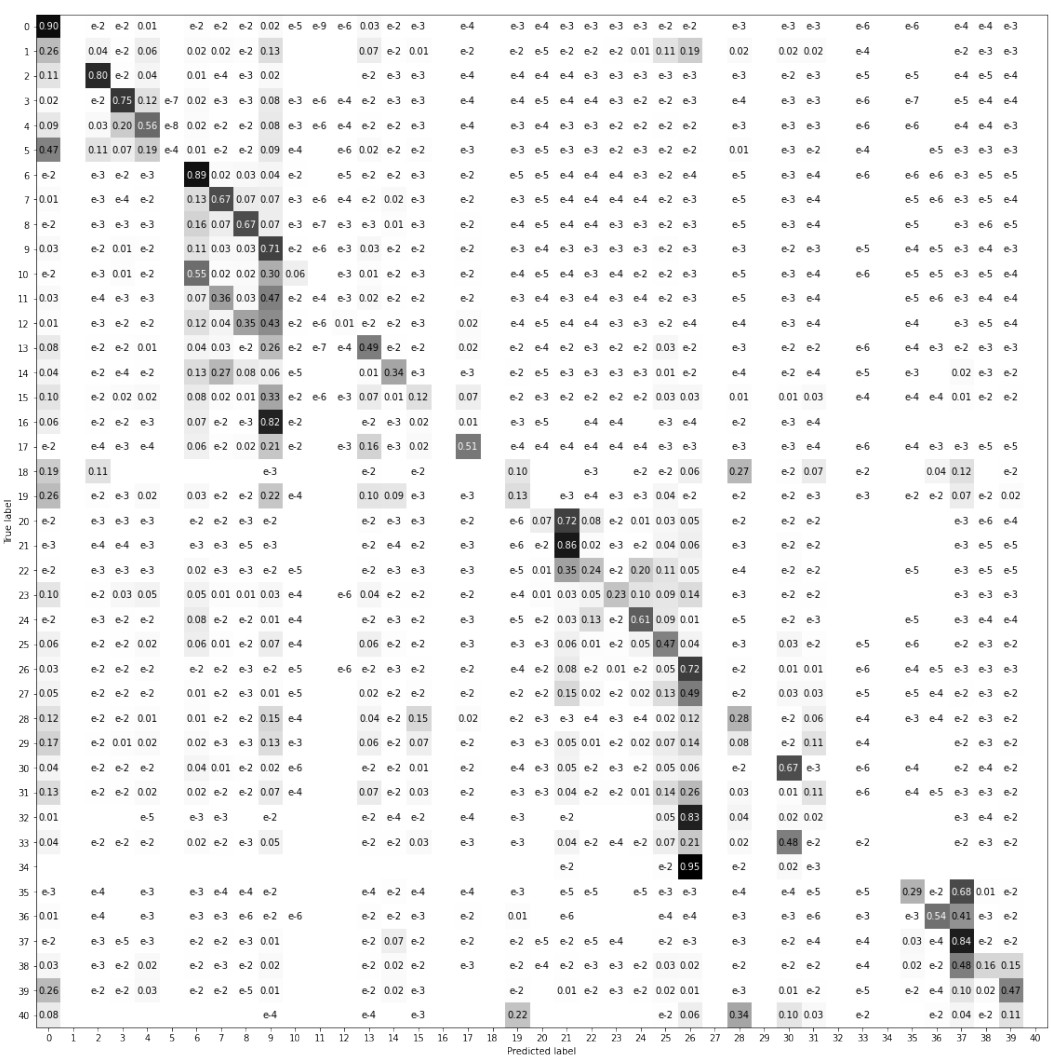

Figure 21: Confusion matrix using the complete CatLC dataset as input data.

