# OpenReview forum: "CatLC: Catalonia Multiresolution Land Cover Dataset"
_NeurIPS.cc/2021/Track/Datasets_and_Benchmarks/Round1 — Submitted to NeurIPS 2021 Datasets and Benchmarks Track (Round 1)_

### Official Review · Reviewer_vgPx · 2021-06-21
**CatLC: Catalonia Multiresolution Land Cover Dataset**

**Rating:** 6
**Confidence:** 4

**Strengths:**

- The dataset seems very interesting. It combines a lot of different sensors and seems to address relevant problems.
- The authors propose a simple baseline method for land cover segmentation.

**Weaknesses:**

- Unfortunately, the paper is poorly written. There are many places, where I was unsure about what was meant. I tried to elaborate on some of issues in the clarity section, but these are just a few examples.
- Evaluation: It seems weird that some of the classes are not present in the test set. How can you evaluate the performance on these classes if they are not in the test set?
- Experiments: It is unclear from the experimental results if the many sensor modalities can improve the predictions and why they are relevant. I undertand that the authors wants the community to figure this out, but I believe that it would be a strong (and rather simple) point to show that multi sensors can improve the segmentation quality over the simple baseline.
- Experiments: Authors motivate two research problems for the dataset. (1) change detection (2) land cover segmentation. They do not give a baseline method for (1) nor explain how the proposed dataset can be used for change detection nor specify what changes users would be interested in.
- Related work is incomplete (see below for more comments)

**Additional Feedback:**

- It would be interesting with some stats on how large each class of pixels are, e.g. there is probably very few pixels with “port”  and a lot of pixels with “Vineyards”.
- Table 2 is difficult to digest. It could be better to show e.g a bar plot. The performance for some classes seems very low. Is this because these classes are very poorly represented in the training set?
- As I understand change detection is also one of the motivations of the paper. I would therefore like to see a simple baseline for change detection also. As I understand you have labels for different seasons, which you could use a GT. You could create a simple baseline as an autoencoder with two encoders (one for spring image, and one for fall image) and one decoder, where information are fused in the bottleneck.
- Another simple, but relevant experiment would be to use the U-Net for different input modalities and see which modality is best for predicting land cover. One would expect the orthophotos as they have highest resolution, but how much worse are the satellite images?

###################

Additional comment after author responds


The authors have answered most of my points. I think the dataset is interesting enough to be accepted, and have therefore changed my vote to marginally accept. I believe especially the method section could be improved to better highlight the relevance of the dataset.

**Clarity:**

- The introduction is poorly written. You can assume that the reader are familiar with CNN, classification and segmentation. I believe that the references to cat/dog does not add  valuable information, but rather confuses the reader. You could simply write that “pixel-wise human annotation of satellite images are challenging and dubious, and thus often requires manual annotation with tools like GSV”.
- I would recommend to delete the current introduction, and use section 4 “Applications” as a new introduction.  Make section 4.1 a separate section called something like “Experiments”.
- “The original images were taken with a resolution of 25 centimeters, but because the land cover map has a resolution of 1 meter, we have decided to rescale the orthophoto raster layer also to 1 meter.” Would it be possible also to release the original resolution. This might become relevant for some applications down the line, e.g. can the land cover map be refined wit the high resolution image?
-  I think the following sentence is difficult to understand: “The slope indicates the steepness of each pixel: The lower its value, the flatter the terrain is”. Does it mean “the pixel intensity/color indicates the terrain’s steepness: yellow means flatter terrain and blue mean more steep terrain”?
- Weird to class this section “Classification” when you do “Segmentation”. These are two different tasks. The goal of classification is to find a label per image, and in segmentation you want to find a label per pixel.
-  “both to have a starting point and to show the most optimal way to train a model with these data”, be careful with stating that it is the most optimal way… if so it limits the use case of the dataset for the research community, but there is probably other ways to train.
- I assume that the input to the U-Net is just the RGB from the Orthophotos? It would be interesting to see what happens if you gave all the different sensor modalities as input to the U-Net? Hopefully the network could learn to use different modalities for different classes, which would be a nice motivation for the dataset. With the current experimental work, it is a bit difficult to see why we need all these sensor modalities?

**Correctness:**

- It seems like beaches, port areas and sea are not in the test set. It is quite strange to have classes in the training set that are not in the test set, as you cannot know how well you perform on these classes if they are not in the test set. I would change the way you create the test/val/train division such that all classes are represented in each set, such that you can evaluate on all performance on the classes.
- It would be good with single number that sum up table 2. Mean IoU could be an idea? This is similar to COCO: https://cocodataset.org/#stuff-eval

**Documentation:**

- As I understand the main intended use case is predicting land cover maps.
- Yes, they include an URL (I would recommend moving this URL from appendix to abstract as many users are more interested in the data/website than the actual paper)

**Ethics:**

- Not to the best of my knowledge.

**Relation To Prior Work:**

-  Seems incomplete. I believe this list of additional arial/satellite datasets would be valuable for the related work section: https://github.com/chrieke/awesome-satellite-imagery-datasets
- It would be beneficial with a table comparing the different datasets especially if the table could make it clear where the CatLC dataset differentiates compared to the other datasets.

**Summary And Contributions:**

The paper proposes a large multi sensor arial dataset consisting of a land cover map (segmentation labels), Orthophoto from airplane photos (RGB + IR), Satellite Photos (Sentinel 1, Sentinel 2) and Topography maps. The dataset covers Catalonia, Spain. The main motivation of the dataset is change detection and land cover predicting, where the authors present a simple baseline model for predicting the latter.

---

> ### Author Response · Authors · 2021-07-15
> **Response to the reviewer vgPx 1/2**
>
> Thank you for the in-depth evaluation of our work and for the constructive feedback. Below we answer the comments and suggestions given in this review.
>
> >Evaluation: It seems weird that some of the classes are not present in the test set. How can you evaluate the performance on these classes if they are not in the test set?
>
> You are right, the previous experiments were not sampled correctly. So, some class was missing in one of the three sets.
> We have redone the sampling. It should be remembered that this dataset does not consist of many images but only one very large image (per layer). So, what we have done is to give a list with the indices of the coordinates (pixel) of the upper left edge for each tile. This list can be found on the dataset web site.
> >Experiments: It is unclear from the experimental results if the many sensor modalities can improve the predictions and why they are relevant. I undertand that the authors wants the community to figure this out, but I believe that it would be a strong (and rather simple) point to show that multi sensors can improve the segmentation quality over the simple baseline.
>
> We have performed a new set of experiments using different data inputs, to demonstrate how using the whole data set and not individual bands improves the results. Section 4 (now called experiments) details this.
>
> >Experiments: Authors motivate two research problems for the dataset. (1) change detection (2) land cover segmentation. They do not give a baseline method for (1) nor explain how the proposed dataset can be used for change detection nor specify what changes users would be interested in.
>
> Our intention was to base this paper exclusively on land cover segmentation, but we have seen that introducing the concept of change detection may have confused the authors as they did not find any experiments on the subject. Therefore, we have eliminated it from the text and it will be left for a future contribution.
>
> >    It seems like beaches, port areas and sea are not in the test set. It is quite strange to have classes in the training set that are not in the test set, as you cannot know how well you perform on these 	classes if they are not in the test set. I would change the way you 	create the test/val/train division such that all classes are represented in each set, such that you can evaluate on all performance on the classes.
>
> This assessment has been answered in a previous question.
>
> >It would be good with single number that sum up table 2. Mean IoU could be an idea? This is similar to COCO: https://cocodataset.org/#stuff-eval
>
> We have decided to convert table 2 into a confusion matrix, which is visually more understandable. It can be found in figure 13 and the Appendix 4: metrics. In addition, we have added the metric mIoU (figures 15 and Appendix 4: metrics).
>
> >The introduction is poorly written. You can assume that the reader are familiar with CNN, classification and segmentation. I believe that the references to cat/dog does not add valuable information, but rather confuses the reader. You could simply write that “pixel-wise human annotation of satellite images are challenging and dubious, and thus often requires manual annotation with tools like GSV”.
> I would recommend to delete the current introduction, and use section 4 “Applications” as a new introduction. Make section 4.1 a separate section called something like “Experiments”.
>
> We have combined the proposals of the three reviewers to modify the introduction. Thus, the first part of chapter 4 has become part of the introduction, and chapter 4 has remained exclusively as experiments. We think it makes the reading more understandable now.
>
> >“The original images were taken with a resolution of 25 centimeters, but because the land cover map has a resolution of 1 meter, we have decided to rescale the orthophoto raster layer also to 1 meter.” Would it be possible also to release the original resolution. This might become relevant for some applications down the line, e.g. can the land cover map be refined wit the high resolution image?
>
> Yes, we have added a section on the website that tells us how to download these images. Keep in mind that an orthophoto at 25cm can be very large, so the download methods are usually a little different, for example using WMS (web map services).
>
> >I think the following sentence is difficult to understand: “The slope indicates the steepness of each pixel: The lower its value, the flatter the terrain is”. Does it mean “the pixel intensity/color indicates the terrain’s steepness: yellow means flatter terrain and blue mean more steep terrain”?
>
> In that case, smaller values relate to blue and higher values to yellow. But it is just a way to represent it here.  We have rewritten the sentence in the hope that it will be better understood.

---

> > ### Author Response · Authors · 2021-07-15
> > **Response to the reviewer vgPx 2/2**
> >
> > >Weird to class this section “Classification” when you do “Segmentation”. These are two different tasks. The goal of classification is to find a label per image, and in segmentation you want to find a label per pixel.
> >
> > You are right, we use classification for this kind of task in the remote sensing area. We have decided to change it to segmentation because we understand it would be more understandable for the neurips community.
> >
> > >“both to have a starting point and to show the most optimal way to train a model with these data”, be careful with stating that it is the most optimal way… if so it limits the use case of the dataset for the research community, but there is probably other ways to train.
> >
> > We meant that AI researchers usually work with multiple images, but there is only one large image in this dataset (per layer). So we have designed a methodology (which is in the tutorial) to make the pipeline easier. We understand your concern about the word optimal, and we have changed it.
> >
> > > I assume that the input to the U-Net is just the RGB from the Orthophotos? It would be interesting 	to see what happens if you gave all the different sensor modalities as input to the U-Net? Hopefully the network could learn to use different modalities for different classes, which would be a nice motivation for the dataset. With the current experimental work, it is a bit difficult to see why we need all these sensor modalities?
> >
> > This assessment has been answered in a previous question.
> >
> > >Seems incomplete. I believe this list of additional arial/satellite datasets would be valuable for the related work section: https://github.com/chrieke/awesome-satellite-imagery-datasets
> > It would be beneficial with a table comparing the different datasets especially if the table could make it clear where the CatLC dataset differentiates compared to the other datasets.
> >
> > Thanks for sharing this resource. We have studied and added one helpful reference. Unfortunately, many of the other landcover datasets are not human-labeled, but by using different machine learning techniques, and others are closed for the moment because they were part of challenges. Nevertheless, we will keep an eye on these and add our dataset to the repository.
> >
> > >Yes, they include an URL (I would recommend moving this URL from appendix to abstract as many users are more interested in the data/website than the actual paper)
> >
> > This is a really good idea, we have added it.
> >
> > >It would be interesting with some stats on how large each class of pixels are, e.g. there is probably very few pixels with “port” and a lot of pixels with “Vineyards”.
> >
> > We have added Figure 3 with the distribution of the classes. Many of them are small compared to others, we could have used a logarithmic axis, but then we would have lost objectivity in the visualization. If you want to know exactly how many pixels there are per class, it is in the Appendix C.
> >
> > >Table 2 is difficult to digest. It could be better to show e.g a bar plot. The performance for some classes seems very low. Is this because these classes are very poorly represented in the training set?
> >
> > We have decided to convert table 2 into a confusion matrix.
> >
> > >As I understand change detection is also one of the motivations of the paper. I would therefore like to see a simple baseline for change detection also. As I understand you have labels for different seasons, which you could use a GT. You could create a simple baseline as an autoencoder with two encoders (one for spring image, and one for fall image) and one decoder, where information are fused in the bottleneck.
> >
> > Change detection was not the motivation of the paper, just a future idea. That is why we have removed it to avoid confusion. However, we keep your idea for future research topics.
> >
> > >Another simple, but relevant experiment would be to use the U-Net for different input modalities and see which modality is best for predicting land cover. One would expect the orthophotos as they have highest resolution, but how much worse are the satellite images?
> >
> > This assessment has been answered in a previous question.

---

### Official Review · Reviewer_4uB8 · 2021-06-30
**Interesting aerial dataset lacking relevant analytical information**

**Rating:** 6
**Confidence:** 3

**Strengths:**

* The dataset can be of interest for communities working with image-based geographical analysis.
* The dataset is made public available with ground truth annotations of 41 labelled classes.
* The dataset consists of multi-modal images, which allow for a more in-depth analysis of the data, compared to e.g., RGB images alone.
* Compared to the presented related work, this dataset covers a significantly larger land area.

**Weaknesses:**

1) Throughout the paper the authors make claims, which I do not find sufficiently backed.
* An example is on line 80-91 where the authors use the size of the land area and the quality and variety of the images as arguments for how suited the dataset is for "AI and deep learning techniques". This is followed by a vague description of the Catalonian landscape and how it is more heterogeneous compared to Alaska and the Saraha desert and therefore better suited for more applications. However,  this is only words and there are no apparent analysis to back up the claims.
* I miss an in-depth analytic evaluation of the proposed dataset, e.g., describing the distribution of each of the classes, such that it is comparable to other similar datasets.
2) The datasets presented in the related work section are described by somewhat varying terms, which makes them difficult to compare. The first two datasets [4] are described by "tiles", "spatial resolution", "bands", and "classes", but the next [5] is presented by "area", "spatial resolution", and "classes". The next is simply "8000 km of roads and four cities", and so on. I would suggest to standardize the description of the individual datasets (as good as possible) , such that they can be more easily and fairly compared to the proposed dataset. E.g., summarizing the findings in a table together with the proposed dataset, thereby providing both an overview of current similar datasets and a clear display of how the proposed dataset stands out.
* I would suggest the authors present the proposed dataset in a manner where it is more easily understandable for computer vision researchers (to make it more accessible). It is extremely difficult to comprehend how many images are in the dataset when only the spatial resolution of 1-10m/px and a total area of 32,000 sq km are provided.
3) The authors state that the "Slope and Aspect" maps (section 3.5.2) are calculated directly from the "Digital Elevation Model". They present no evaluation or discussion of how this additional information can be used to enhance performance of some arbitrary method. In other words, the "Slope and Aspect" maps contain redundant information as it is already available through the "Digital Elevation Model" images.
4) In section "Classification" (section 4.1, line 226-227) the authors state that they "... show the most optimal way to train a model with these data.", but they simply train a public available segmentation network and measure the performance, which has nothing to do with how to train a model in the most optimal way.
5) In section "Classification" (section 4.1) the authors provide an "initial benchmark". However, the authors state that the dataset is randomly divided into train, validation, and test splits. Are these splits published? Otherwise it can hardly be considered a benchmark since it cannot be replicated.
* Furthermore, the authors only use the RGB orthophotos and simply ignore the rest of the dataset. The evaluation could have been more informative if it was conducted in a way where the pros and cons of the various sensor types were illustrated.
* It is difficult to get anything from the results in table 2, when there are no information on the distribution of the classes. How are the classes distributed in the three splits? How are they distributed overall? Do we see overfitting on some of the classes?

**Additional Feedback:**

I find the proposed dataset both interesting and relevant and it is my impression that the authors have put a lot of time and effort into constructing it.
However, I find it problematic that there is (more or less) no analysis of the dataset. E.g., how are the 41 classes distributed, how do the sensors support each other, etc.
I don't find it suitable for NeurIPS in it's current state.


** Comments after reading the second and updated version of the paper:
The authors have improved several parts of the paper based on the comments from the reviewers.
The analytic parts of the paper could still be improved, however, I choose to put more weight on the dataset itself, which I think could be interesting and relevant for the community. Based on the improvements I have changed my rating from "rejection" to "marginally above acceptance threshold".

**Clarity:**

To some degree. The language is understandable, but there are several claims, which are not backed by proof. See my comments  here and under "weaknesses" for examples.

1) I suggest to make it clear that the three figures in Figure 4, Figure 5, Figure 6, Figure 7, Figure 8, Figure 9, and Figure 10 are of the same area.
* Also, caption sub-figures (throughout the entire paper), such that it is possible to reference the individual sub-figures.
2) Remember dots at the end of captions.
3) I find the descriptions of the sensors/dataset parts (section 3.1 -> 3.5) of varying quality.
* In section 3.5 "Topographic Maps" it says that there are five different topographic maps, but they are described in 4 subsections, which may be a bit confusing.
* Why are there no image examples of 3.5.2 and 3.5.3?
* Line 144-145 could the VH polarization provide additional valuable information? The dataset is already 1.4 TB, so the reasoning of not taking it into account in order to minimize the amount of information seems a bit odd.
* Line 149-162 is better suited for supplementary material in my opinion.
* Line 171-179 are also for the suppl. mat. in my opinion. I would also suggest to present images before and after they have been corrected, otherwise it is difficult to comprehend for people not familiar with e.g., level-2a images.




**Correctness:**

To some degree.
Several claims may be correct, but is not backed up by sufficient scientific proof. See my comments.


**Documentation:**

See my other comments.

**Ethics:**

I don't think so.

**Relation To Prior Work:**

I think the authors could make it more clear how their work differs from previous published datasets. See my comments in "Weaknesses"

**Summary And Contributions:**

The paper describes and presents a public available aerial multi-modal image-based dataset of the Spanish region of Catalonia.
The dataset is labelled with a total of 41 classes from four main topics covering agriculture, forest, urban areas, and water bodies.
The authors evaluate the performance of a segmentation network on the RGB orthophotos from the dataset.

---

> ### Author Response · Authors · 2021-07-15
> **Response to the reviewer 4uB8 1/2**
>
> Thank you for the in-depth evaluation of our work and for the constructive feedback. Below we answer the comments and suggestions given in this review.
>
> >An example is on line 80-91 where the authors use the size of the land area and the quality and variety of the images as arguments for how suited the dataset is for "AI and deep learning techniques". This is followed by a vague description of the Catalonian landscape and how it is more heterogeneous compared to Alaska and the Saraha desert and therefore better suited for more applications. However, this is only words and there are no apparent analysis to back up the claims.
>
> You are right. We understand that this statement is not very scientific. So, as I have commented to the previous reviewer, we have entirely changed the introduction. We have based it on the beginning of the previous chapter 4, leaving chapter 4 only about experiments.
>
> >I miss an in-depth analytic evaluation of the proposed dataset, e.g., describing the distribution of each of the classes, such that it is comparable to other similar datasets.
>
> We have added table 3 that will help in the description of the proposed dataset. Also, there is figure 3, where we can find the distribution for each land cover class. If the reader wants to enter in more detail, in Appendix 3, there is also the distribution for the train/evaluation/test sets.
>
> >The datasets presented in the related work section are described by somewhat varying terms, which makes them difficult to compare. The first two datasets [4] are described by "tiles", "spatial resolution", "bands", and "classes", but the next [5] is presented by "area", "spatial resolution", and "classes". The next is simply "8000 km of roads and four cities", and so on. I would suggest to standardize the description of the individual datasets (as good as possible) , such that they can be more easily and fairly compared to the proposed dataset. E.g., summarizing the findings in a table together with the proposed dataset, thereby providing both an overview of current similar datasets and a clear display of how the proposed dataset stands out.
>
> The suggested overview can be found in Table 1.
>
> >I would suggest the authors present the proposed dataset in a manner where it is more easily understandable for computer vision researchers (to make it more accessible). It is extremely difficult to comprehend how many images are in the dataset when only the spatial resolution of 1-10m/px and a total area of 32,000 sq km are provided.
>
> As said before, in the table 3, we summarized the proposed dataset.
>
> >The authors state that the "Slope and Aspect" maps (section 3.5.2) are calculated directly from the "Digital Elevation Model". They present no evaluation or discussion of how this additional information can be used to enhance performance of some arbitrary method. In other words, the "Slope and Aspect" maps contain redundant information as it is already available through the "Digital Elevation Model" images.
>
> It is true, they are redundant, and maybe they cannot enhance performance, but we think they might be helpful to interpret the results better. So, in order to make it more straightforward, section 3.5 only contains three topographic maps, and inside the first one, we explain the aspect and the slope.
>
> >In section "Classification" (section 4.1, line 226-227) the authors state that they "... show the most optimal way to train a model with these data.", but they simply train a public available segmentation network and measure the performance, which has nothing to do with how to train a model in the most optimal way.
>
> AI researchers usually work with multiple images, but there is only one large image in this dataset (per layer). So we have designed a methodology (which is in the tutorial) to make the pipeline easier. We understand your concern about the word optimal, and we have changed it. So we were not talking about the model being optimal, but the pipeline.
>
> >In section "Classification" (section 4.1) the authors provide an "initial benchmark". However, the authors state that the dataset is randomly divided into train, validation, and test splits. Are these splits published? Otherwise it can hardly be considered a benchmark since it cannot be replicated.
>
> The slipts are now published in the dataset webpage.
>
> >Furthermore, the authors only use the RGB orthophotos and simply ignore the rest of the dataset. The evaluation could have been more informative if it was conducted in a way where the pros and cons of the various sensor types were illustrated.
>
> You are correct that we had been too brief in using only the orthophotos. So, chapter 4 has been rewritten again with more experiments comparing the use of different sensors.

---

> > ### Author Response · Authors · 2021-07-15
> > **Response to the reviewer 4uB8 2/2**
> >
> > >It is difficult to get anything from the results in table 2, when there are no information on the distribution of the classes. How are the classes distributed in the three splits? How are they distributed overall? Do we see overfitting on some of the classes?
> >
> > Table 2 has been converted into a confusion matrix for each of the experiments. We have also added a comparison using the metrics recommended by the coco dataset of the mean intersection over union. Also, we have added the distribution for the three splits.
> >
> > > 1. I suggest to make it clear that the three figures in Figure 4, Figure 5, Figure 6, Figure 7, Figure 8, Figure 9, and Figure 10 are of the same area. Also, caption sub-figures (throughout the entire paper), such that it is possible to reference the individual sub-figures.
> > 2. Remember dots at the end of captions.
> > 3. I find the descriptions of the sensors/dataset parts (section 3.1 -> 3.5) of varying quality.
> >
> > It is now fixed.
> >
> > >In section 3.5 "Topographic Maps" it says that there are five different topographic maps, but they are described in 4 subsections, which may be a bit confusing.
> >
> > This section has been rewritten to clarify the characteristics of the different topographic data, highlighting those that are most important.
> >
> > >Why are there no image examples of 3.5.2 and 3.5.3?
> >
> > It has been solved with the new version of section 3.5.
> >
> > >Line 144-145 could the VH polarization provide additional valuable information? The dataset is already 1.4 TB, so the reasoning of not taking it into account in order to minimize the amount of information seems a bit odd.
> >
> > It is not only a matter of the amount of information, but we also believe that it will not provide significant added value. In any case, its use can be evaluated more accurately. These comments have been added to the text.
> >
> > >Line 149-162 is better suited for supplementary material in my opinion.
> > Line 171-179 are also for the suppl. mat. in my opinion. I would also suggest to present images before and after they have been corrected, otherwise it is difficult to comprehend for people not familiar with e.g., level-2a images.
> >
> > We have followed your suggestions. We hope it is more clear now.
> >
> > >I find the proposed dataset both interesting and relevant and it is my impression that the authors have put a lot of time and effort into constructing it. However, I find it problematic that there is (more or less) no analysis of the dataset. E.g., how are the 41 classes distributed, how do the sensors support each other, etc. I don't find it suitable for NeurIPS in it's current state.
> >
> > The analysis of the data has been extended, and more experiments have been done. It can be seen in Chapters 3 and 4.

---

> > > ### Comment · Reviewer_4uB8 · 2021-07-20
> > > **New rating**
> > >
> > > @Authors: Thank you for taking your time to respond to my questions and comments and for updating the paper accordingly.
> > >
> > > @Chair: The authors have improved several parts of the paper based on the comments from the reviewers. The analytic parts of the paper could still be improved, however, I choose to put more weight on the dataset itself, which I think could be interesting and relevant for the community. Based on the improvements I have changed my rating from "rejection" to "marginally above acceptance threshold".

---

### Official Review · Reviewer_F31g · 2021-07-05
**Review of the NeurIPS 2021 Track Datasets and Benchmarks Submission "CatLC: Catalonia Multiresolution Land Cover Dataset**

**Rating:** 7
**Confidence:** 3

**Strengths:**

- The CatLC data set indeed provides a novel, labelled collection of remote sensing images of unprecedented area coverage, directly available multi-layers and high resolution of 1 m, which can be used in the development of machine learning models for large scale computer vision tasks.

- It seems, after checking the provided tutorials and code, that the data set has indeed been curated in such a way that it can be mostly readily used, also by a non-expert in specific remote-sensing techniques. It thus should be directly usable for the broader computer vision community for development.

- While the main focus of CatLC seems to be to foster the development of models for land cover map segmentation, the range of different layers provided in the data set open up opportunities to explore sensor fusion techniques or focus on different aspects of the problem.

**Weaknesses:**

- It is not entirely clear to the reviewer what the exact aims of publishing the CatLC data set in this fashion are. It is clear that the general goal is to foster the development of deep learning based image segmentation methods for remote sensing. However some of the more specific goals are not clearly collected in one place in the text, such as whether it is to improve the automated labelling of land cover maps or to generally stimulating research and development, or whether the main audience is supposed to be general computer vision experts interested in remote sensing ,or remote sensing experts with knowledge in computer vision.

- In addition, the first benchmark study is very short and only using one type of images in the data set, which raises the question what the exact goals are for the specific combination of data types found in CatLC.

- Unfortunately, the main part of the paper - the description of the CatLC data set itself - has some weaknesses in presentation, especially of details in the data, which could also be remedied by general better documentation (see below). A few major examples:
    * The land cover map containing the ground truth labels is not well explained, at least to a non-expert in land cover   remote sensing. How excatly have the original labels been obtained? What was the exact procedures used to work out the changes from 2009 to 2018, and is the 81% thematic accuracy obtained in the test sample a good value in this field?

    * The Orthophoto data set is described to brief and without enough detail. How many images at what sizes are available? What are the digital make-up tasks performed? Why the 1% percentage threshold?

    * For the sentinel-1 data, the reason struck the reviewer as odd that the authors suggested for why they left out part of the data because it would be better to reduce the information. Since CatLC is essentially a collection of different sub data sets, why not let the user decide how much information to use?

- The benchmark study is very short and there is not enough room given for proper descriptions of the procedures and results (which may be just a presentation issue, since the code  - very briefly reviewed - seems quite solid). A range of questions are a bit unclear and left open for the reader:

    * How was the evaluation done exactly? Has there been cross-validation procedures used? What was the validation data used for? Why not also display the accuracy?

    * It also is a bit unclear how the results of that first benchmark are considered by the authors themselves. How do the authors judge the highly different performance over class labels, for instance? Is that a clear results of scarcity of some labels? Further, a comparison with typical results for this tasks in land cover maps is completely missing thus rendering the worth of the benchmark study a bit unclear.

    * It is a bit misleading to consider this a benchmark of the whole data set, since essentially this is just a study on one part - the orthophoto data.

    * It would have been nice to see at least some discussion of possibilities here, e.g. other architectures beyond the U-net, or how to make use of the full data set together.

- While the data set has indeed been curated by remote sensing experts for a more general audience with tutorials and code are provided, the presentation in the paper is sometimes still very focused on specific terminology and techniques from remote sensing and thus not always easy to follow for a non-expert in that field, for instance in Section 3.3.

**Additional Feedback:**

 A specific question that the reviewer was wondering about is whether the random assignment of tiles over all classes to train, test and validation data in the benchmark study is really preserving the i.i.d. principle since there are probably quite some correlation between nearby regions in the map?

**** Final Comment after paper revision ****

The authors addressed many of the concerns raised in my review in their revised version. While I still think that there would be still more potential in presentation and analytical analysis of the data and benchmark study i clearly put my emphasis on the data set and the clear effort the authors put into collecting and presenting it to the wider computer vision community. I changed the rating accordingly to an accept.

**Clarity:**

There are some problems with clarity and structure in the paper that could be easily addressed and are thus not a major issue. Main examples:

- sometimes there are wording and grammar issues hindering the clarity of statements, e.g. in line 27-28, or 144 the use of the word "open" (has been considered? Open to the user?).

- Image captions could be improved. For instance, it was quite unclear at first glance that all pictures cover the same area.

- The aforementioned lack in clarity of the exact goals and aims of the CatLC data set. A short and concise bullet list in the introduction could be helpful, for example.


**Correctness:**

To the best of the knowledge and understanding of the reviewer the basic construction of the data set is fine and correct. Also the benchmark study itself seems to have been conducted correctly on the technical level judging by a brief overview of the code. The issues of the reviewer are more with presentation and discussion (see weaknesses and documentation).

**Documentation:**

- In general, a unified description of the data set with all ist parts and parameters is missing, listing exactly all the features of the different data layers, also including more technical elements together in one place (number of images, size of the data sets, statistics over labelled regions etc.). This could be either in form of a collection of tables, or as supplemental data. This seems also be true for the additional material at the provided links. At the very least, this type of information is not well collected in one place. There could also be a language problem of the reviewer, because some of the material is only available in Catalan (such as documentation of the land cover map).

- In this context, the authors could take more care in describing the used third party data e.g. from ESA and ICGC, and especially the type of preprocessing steps to be expected in this data, since this is one of the central problems of data aggregation.

**Ethics:**

No, the reviewer agrees with the author's assessment given in the paper checklist.

**Relation To Prior Work:**

The paper contains a section on previously published similar labelled land cover data sets and mentions the main features of those data sets. While the reader can compare the characteristics themselves, a clear comparison of the CatLC data set to these previous sets by the authors is lacking in the rest of the paper

**Summary And Contributions:**

This submission presents a new remote sensing data set - the "Catalonia Multiresolution Land Cover Dataset (CatLC)" -  composed of images from aircraft, satellites and from existing topographic maps of the region of Catalonia. The aim is to provide a data set with good resolution covering a large and heterogeneous area to the research community to foster the development of deep learning based image segmentation models for remote sensing, especially for land cover maps. The data set, along with tutorials, code and descriptions is provided at a number of links hosted by the Institut Cartografic i Geologic de Catalunya.

---

> ### Author Response · Authors · 2021-07-15
> **Response to the reviewer F31g 1/2**
>
> Thank you for carefully examining our work. We provide below an answer to the questions raised in the review.
>
> >It is not entirely clear to the reviewer what the exact aims of publishing the CatLC data set in this fashion are. It is clear that the general goal is to foster the development of deep learning based image segmentation methods for remote sensing. However some of the more specific goals are not clearly collected in one place in the text, such as whether it is to improve the automated labelling of land cover maps or to generally stimulating research and development, or whether the main audience is supposed to be general computer vision experts interested in remote sensing ,or remote sensing experts with knowledge in computer vision.
>
> The first part of chapter 4 has become part of the introduction, and chapter 4 has remained exclusively as experiments. We have added your recommendations and tried to be more concise on the objectives.
>
> >In addition, the first benchmark study is very short and only using one type of images in the data set, which raises the question what the exact goals are for the specific combination of data types found in CatLC.
>
> We have performed a new set of experiments to better benchmark, using different data inputs, to demonstrate how using the whole data set and not individual bands improves the results. Section 4 (now called experiments) details this.
>
> >The land cover map containing the ground truth labels is not well explained, at least to a non-expert in land cover remote sensing. How excatly have the original labels been obtained? What was the exact procedures used to work out the changes from 2009 to 2018, and is the 81% thematic accuracy obtained in the test sample a good value in this field?
>
> The paper's objective is not to go into details of the elaboration of the land cover map since the length of the paper is limited. Nevertheless, in the dataset webpage, under the metadata, there is information about the process. We are sorry that before, the link was linked to a Catalan site, now it is fixed and links to the English version. In any case, it should be noted that the land cover map presented has been generated by a prestigious institution in the world of cartography, such as the ICGC, which offers a guarantee of quality with current standards.
>
> >The Orthophoto data set is described to brief and without enough detail. How many images at what sizes are available? What are the digital make-up tasks performed? Why the 1% percentage threshold?
>
> It should be noted that the repository contains only one huge image (the entire surface of Catalonia) per layer. To work with them, we have done tiles on the fly (throughout the training, we have designed a specific pipeline for this). So we understand that for the neruips community, it is more clarifying the explanation that we have added in the new table 3 with a summary of how the tiles are organized. The more technical details of its elaboration can be found in the same link of the previous comment.
>
> >For the sentinel-1 data, the reason struck the reviewer as odd that the authors suggested for why they left out part of the data because it would be better to reduce the information. Since CatLC is essentially a collection of different sub data sets, why not let the user decide how much information to use?
>
> In this case, we believe that the additional information may be redundant. Therefore it has not been included in the dataset. In any case, users are left open to use this supplementary information. They can download and process as we explain in the Sentinel-1 sections.
>
> >The benchmark study is very short and there is not enough room given for proper descriptions of the procedures and results (which may be just a presentation issue, since the code - very briefly reviewed - seems quite solid).
>
> We have performed a new set of experiments using different data inputs to demonstrate how using the whole data set and not individual bands improves the results. Section 4 (now called experiments) details this.
>
> >While the data set has indeed been curated by remote sensing experts for a more general audience with tutorials and code are provided, the presentation in the paper is sometimes still very focused on specific terminology and techniques from remote sensing and thus not always easy to follow for a non-expert in that field, for instance in Section 3.3.
>
> We have moved sentinel-1 and sentinel-2 technical procedures to the supplementary material as we agree it is not easy to follow for non-experts in remote sensing.
>
> >There are some problems with clarity and structure in the paper that could be easily addressed and are thus not a major issue.
>
> We have fixed them, we hope it is more clear now.
>
> >The aforementioned lack in clarity of the exact goals and aims of the CatLC data set.
>
> The introduction has been redone completetly, we hope that now it is more clear.

---

> > ### Author Response · Authors · 2021-07-15
> > **Response to the reviewer F31g 2/2**
> >
> > >In general, a unified description of the data set with all ist parts and parameters is missing, listing exactly all the features of the different data layers, also including more technical elements together in one place (number of images, size of the data sets, statistics over labelled regions etc.). This could be either in form of a collection of tables, or as supplemental data. This seems also be true for the additional material at the provided links. At the very least, this type of information is not well collected in one place. There could also be a language problem of the reviewer, because some of the material is only available in Catalan (such as documentation of the land cover map).
> > In this context, the authors could take more care in describing the used third party data e.g. from ESA and ICGC, and especially the type of preprocessing steps to be expected in this data, since this is one of the central problems of data aggregation.
> >
> > We have re-organized the paper, including new tables (for the data and the experiments) and graphs to facilitate the understanding of the paper. We have also fixed the link to the Catalan site (now goes to an English version).
> >
> > >A specific question that the reviewer was wondering about is whether the random assignment of tiles over all classes to train, test and validation data in the benchmark study is really preserving the i.i.d. principle since there are probably quite some correlation between nearby regions in the map?
> >
> > A new random assignment of the tiles has been set. Being this a segmentation problem makes it very difficult to distribute all the classes identically. We have, by trial and error, tried to have the dataset as balanced as possible. There is a new figure in  Appendix 3 with the distribution of the data.

---

### Author Response · Authors · 2021-07-15
**Response to all reviewers**

We want to thank all the reviewers for their time and feedback. Please find the answers to the raised questions/concerns below. We have also updated the paper and used blue color to highlight the new sections added.

The introduction has been changed almost completely, bringing part from the previous "applications" section. We have tried to solve the problems that have to do with the clarity of the manuscript.

**Data information**
We have added table 1 to make more clear the comparison between the commented datasets.
We have added table 3 to understand in one sight of an eye how the details from CatLC dataset.
We have edited many parts of the chapter that describe the dataset to make it now more comprehensible.
We have moved information about the preprocessing of Sentinel-1 and Sentinel-2 to the appendix. We have also added a pair of images showing the correction of Sentinel-2 images.

**Data distribution**
We have added figure 3 with the distribution of the data. Also, in Appendix 3, we can find figure 17 with the distribution of the dataset in the three train/evaluation/test sets.

**Experiments**
We have not only kept the orthophoto experiments, but we have also added sentinel-2 and the complete dataset. Section 4 has been completely redesigned, changing the title from applications to experiments.
New figures 13, 15, 18, 19, 20, 21 accompany the results.

---

### Decision · Program_Chairs · 2021-07-27

**Decision:**

Reject

**Comment:**

The paper presents a new land cover dataset comprising aerial images of Catalonia together with a number of semantic and geographic annotations. All reviewers felt that the dataset was interesting and likely to be useful; its scale is particularly impressive, as it covers the entirety of Catalonia.

The main weaknesses identified by all reviewers have to do with the clarity of the writing and presentation in the paper. All reviewers gave detailed examples of unclear passages, and suggested a large number of edits and improvements. The authors implemented many of these suggestions in their revised draft, which caused all reviewers to raise their scores to recommend acceptance. In their final comments, all reviewers mentioned that their final rating was ultimately weighted more toward the utility of the dataset, but that the paper itself still needs improvement.

After reviewing the reviewer comments and the paper, the AC agrees that the dataset itself is novel, interesting, impressive, and worthy of publication. The AC was impressed by the willingness of the authors to incorporate feedback from the reviewers to improve the paper; however even after revision the quality of presentation and writing is not up to the standard we expect from NeurIPS publications.

Therefore after consulting with the Program Chairs, we have decided to reject the paper for now. However we emphasize that this decision is solely based on the clarity and presentation of the written paper; the dataset itself is of high quality and is exactly the sort of contribution that we intend to publish in this track. We therefore encourage the authors to make another round of revision and rewriting to the paper, then resubmit to the 2nd round of the Datasets and Benchmarks Track at the end of August.

The reviewers have already given great suggestions for improving the paper. The AC would like to echo the recommendation of making the paper more accessible to readers familiar with computer vision and machine learning, but not land cover or remote sensing. Some of Figures 4-12 might be slightly shrunk and combined in order to give more space for additional definitions, explanations, or diagrams that are currently relegated to the supplementary material or website.

We understand that this may not be the outcome that authors expected. However we truly believe that more clearly presenting and describing the dataset for the NeurIPS community will ultimately lead to more users, and lead to more impact overall in the long run.